# Immunofluorescence Analysis as a Diagnostic Tool in a Spanish Cohort of Patients with Suspected Primary Ciliary Dyskinesia

**DOI:** 10.3390/jcm9113603

**Published:** 2020-11-09

**Authors:** Noelia Baz-Redón, Sandra Rovira-Amigo, Mónica Fernández-Cancio, Silvia Castillo-Corullón, Maria Cols, M. Araceli Caballero-Rabasco, Óscar Asensio, Carlos Martín de Vicente, Maria del Mar Martínez-Colls, Alba Torrent-Vernetta, Inés de Mir-Messa, Silvia Gartner, Ignacio Iglesias-Serrano, Ana Díez-Izquierdo, Eva Polverino, Esther Amengual-Pieras, Rosanel Amaro-Rodríguez, Montserrat Vendrell, Marta Mumany, María Teresa Pascual-Sánchez, Belén Pérez-Dueñas, Ana Reula, Amparo Escribano, Francisco Dasí, Miguel Armengot-Carceller, Marta Garrido-Pontnou, Núria Camats-Tarruella, Antonio Moreno-Galdó

**Affiliations:** 1Vall d’Hebron Institut de Recerca (VHIR), Vall d’Hebron Hospital Universitari, Vall d’Hebron Barcelona Hospital Campus, 08035 Barcelona, Spain; noelia.baz@vhir.org (N.B.-R.); srovi@yahoo.es (S.R.-A.); mfcancio75@gmail.com (M.F.-C.); atorrentvernetta@gmail.com (A.T.-V.); idemir@vhebron.net (I.d.M.-M.); silviagartner@gmail.com (S.G.); nachotela@gmail.com (I.I.-S.); anadiezizquierdo@gmail.com (A.D.-I.); evapo74@gmail.com (E.P.); belen.perez@vhir.org (B.P.-D.); nuria.camats@vhir.org (N.C.-T.); 2Department of Pediatrics, Obstetrics, Gynecology, Preventative Medicine and Public Health. Universitat Autònoma de Barcelona, 08193 Barcelona, Spain; 3Department of Pediatrics, Vall d’Hebron Hospital Universitari, Vall d’Hebron Barcelona Hospital Campus, 08035 Barcelona, Spain; 4CIBER of Rare Diseases (CIBERER), Instituto de Salud Carlos III (ISCIII), 28029 Madrid, Spain; 5Pediatric Pulmonology Unit, Hospital Clínico Universitario de Valencia, 46010 Valencia, Spain; castillo_sil@gva.es (S.C.-C.); aescribano@separ.es (A.E.); 6Pediatric Pulmonology Department and Cystic Fibrosis Unit, Hospital Sant Joan de Déu, 08950 Barcelona, Spain; MCols@sjdhospitalbarcelona.org; 7Pediatric Pulmonology Unit, Department of Pediatrics, Hospital del Mar, 08003 Barcelona, Spain; MACaballeroRabasco@parcdesalutmar.cat; 8Pediatric Pulmonology Unit, Hospital Parc Taulí de Sabadell, 08208 Sabadell, Spain; oasensio58@gmail.com; 9Pediatric Pulmonology Unit, Hospital Miguel Servet, 50009 Zaragoza, Spain; carl_zaragoza@yahoo.es; 10Pediatric Pulmonology Unit, Hospital Germans Trias i Pujol, 08916 Badalona, Spain; mimarmartinez@gmail.com; 11Pneumology Department, Vall d’Hebron Hospital Universitari, Vall d’Hebron Barcelona Hospital Campus, 08035 Barcelona, Spain; 12Department of Pediatrics, Hospital Universitario Son Llàtzer, 07198 Palma de Mallorca, Spain; eamengua@hsll.es; 13Pneumology Department, Hospital Clínic, 08036 Barcelona, Spain; ramaro@clinic.cat; 14Pneumology Department, Hospital Josep Trueta, 17007 Girona, Spain; mvendrell.girona.ics@gencat.cat; 15Girona Biomedical Research Institute (IDIBGI), Universitat de Girona, 17190 Girona, Spain; 16CIBER of Respiratory Diseases (CIBERES), Instituto de Salud Carlos III (ISCIII), 28029 Madrid, Spain; miguel.armengot@gmail.com; 17Pediatric Pulmonology Unit, Consorci Sanitari de Terrassa, 08191 Terrassa, Spain; 39769mme@comb.cat; 18Pediatric Pulmonology Unit, Hospital Universitari Sant Joan de Reus, 43204 Tarragona, Spain; maitepasc@gmail.com; 19Grupo de Biomedicina Molecular, Celular y Genómica, IIS La Fe, 46026 Valencia, Spain; areumar91@gmail.com; 20Department of Paediatrics, Obstetrics and Gynecology, Universitat de Valencia, 46010 Valencia, Spain; Dasi@uv.es; 21UCIM, Rare Respiratory Diseases Research Group, Instituto de Investigación Sanitaria INCLIVA, 46010 Valencia, Spain; 22Otorhinolaryngology Department, Hospital Universitario y Politécnico La Fe, 46026 Valencia, Spain; 23Department of Pathology, Vall d’Hebron Hospital Universitari, Vall d’Hebron Barcelona Hospital Campus, 08035 Barcelona, Spain; magarrido@vhebron.net

**Keywords:** cilia, primary ciliary dyskinesia, PCD, immunofluorescence, antibody

## Abstract

Primary ciliary dyskinesia (PCD) is an autosomal recessive rare disease caused by an alteration of ciliary structure. Immunofluorescence, consisting in the detection of the presence and distribution of cilia proteins in human respiratory cells by fluorescence, has been recently proposed as a technique to improve understanding of disease-causing genes and diagnosis rate in PCD. The objective of this study is to determine the accuracy of a panel of four fluorescently labeled antibodies (DNAH5, DNALI1, GAS8 and RSPH4A or RSPH9) as a PCD diagnostic tool in the absence of transmission electron microscopy analysis. The panel was tested in nasal brushing samples of 74 patients with clinical suspicion of PCD. Sixty-eight (91.9%) patients were evaluable for all tested antibodies. Thirty-three cases (44.6%) presented an absence or mislocation of protein in the ciliary axoneme (15 absent and 3 proximal distribution of DNAH5 in the ciliary axoneme, 3 absent DNAH5 and DNALI1, 7 absent DNALI1 and cytoplasmatic localization of GAS8, 1 absent GAS8, 3 absent RSPH9 and 1 absent RSPH4A). Fifteen patients had confirmed or highly likely PCD but normal immunofluorescence results (68.8% sensitivity and 100% specificity). In conclusion, immunofluorescence analysis is a quick, available, low-cost and reliable diagnostic test for PCD, although it cannot be used as a standalone test.

## 1. Introduction

Primary ciliary dyskinesia (PCD) is an autosomal recessive rare disease (1/15,000) caused by an alteration of ciliary structure, which impairs mucociliary clearance [1,2]. Symptoms of PCD may include persistent wet cough from early infancy, recurrent respiratory infections, bronchiectasis, chronic rhinosinusitis, persistent otitis media with effusion and associated conductive hearing loss, male infertility, female subfertility, situs inversus in half of PCD patients [1,2,3] and heterotaxic defects in 6–12% of cases [4].

Diagnosis is often delayed, with the possibility of an impairment of lung function [1], because of non-specificity of PCD symptoms and limitations of the available techniques [5]. According to the European Respiratory Society (ERS), PCD diagnosis is commonly based on studying the ciliary function by high-speed video-microscopy (HSVM) and ciliary ultrastructure by transmission electron microscopy (TEM) [6]. As there is not a unique gold standard diagnostic test, the ERS [6] and the American Thoracic Society (ATS) [7] have recently proposed the use of different diagnostic techniques to improve the accuracy and diagnosis rate of PCD.

Immunofluorescence (IF) is a technique consisting in the use of fluorescently labeled antibodies for the detection of the presence and distribution of different ciliary proteins in human respiratory cells by fluorescence or confocal microscopy, and it has been recently proposed as a tool to improve the diagnosis rate in PCD and facilitate a better understanding of disease-causing genes [6,7]. Motile respiratory cilia (9 + 2 cilia) are composed of nine peripheral microtubular doublets (composed of A and B tubules) and a central pair (C1 and C2) surrounded by a protein central sheath. An important number of protein complexes are distributed among these microtubule structures: the outer (ODA) and inner dynein arms (IDA), the nexin links, the central sheath and the radial spokes [8,9] (Figure 1). The methodology for IF staining of ciliated respiratory epithelial cells was first described by Omran and Loges [10]. Nowadays, an important number of antibodies against different cilia proteins are available, including antibodies against proteins in ODA, IDA, radial spoke head and dynein regulatory complex [6]. IF is cheaper and easier than other techniques and the use of IF as a diagnostic test in PCD is likely to increase as more antibodies become available [6]. However, studies on the use of IF in diagnostic settings and IF validation studies are necessary to consider IF as a diagnostic tool for PCD in diagnostic cohort studies [6]. The ATS considers the IF as one of the emerging PCD diagnosis techniques, although it has emphasized the lack of consensus on a gold standard for diagnosis and the insufficient sensitivity and specificity when applied to the general population [7].

A study by Shoemark et al. demonstrated that IF is a useful diagnostic technique and presents the same accuracy as well-performed TEM analysis, which is why the authors support IF as a routine diagnostic test for PCD, especially when TEM expertise or equipment is not available [5]. To our knowledge, this is the only study evaluating the accuracy of conventional IF for the diagnosis of PCD. In a recent study, Liu et al. presented a quantitative super-resolution imaging workflow for the detection of cilia defects thanks to the validation of 21 commercially available IF antibodies. Molecular defects using this super-resolution imaging toolbox were described in 31 clinical PCD cases, including patients with negative TEM results and/or with genetic variants of uncertain significance (VUS) [11].

We hypothesized that an IF panel would be a useful technique to study the cilia structure and improve PCD diagnosis, especially in settings with low availability of TEM results. Our aim was to establish the utility and accuracy for PCD diagnosis of an IF panel in a Spanish cohort of patients with suspected PCD in relation to clinical characteristics, genetics and/or HSVM.

## 2. Experimental Section

### 2.1. Patients

This study belongs to a prospective multicentric study including all 74 consecutive patients with a clinical history suggestive of PCD during the period from 2016 to 2020.

This project was approved by the Clinical Research Ethics Committee (CEIC) of Hospital Universitari Vall d’Hebron (PR(AMI)148/2016). Written informed consent was obtained from ≥18-year-old patients, from ≥12-year-old patients and their parents or guardians and from <12-year-old patients’ parents or guardians.

The majority of patients attended the Hospital Universitari Vall d’Hebron (HUVH), and patients and samples from other hospitals from Spain were also analyzed: Hospital Sant Joan de Déu (Esplugues de Llobregat, Barcelona), Hospital del Mar (Barcelona), Hospital Parc Taulí (Sabadell, Barcelona), Hospital Germans Trias i Pujol (Badalona, Barcelona), Hospital Clínic (Barcelona), Hospital Miguel Servet (Zaragoza), Hospital Josep Trueta (Girona), Hospital Universitari Sant Joan de Reus (Reus, Tarragona), Consorci Sanitari de Terrassa (Terrasa, Barcelona) and PCD group Valencia (Hospital Universitario y Politécnico la Fe, Hospital Clínico Universitario de Valencia and INCLIVA).

### 2.2. PCD Diagnostic Evaluation

Patients were evaluated for PCD with a clinical symptoms questionnaire and PICADAR (PrImary CiliARy DyskinesiA Rule) score [12], nasal nitric oxide (nNO), high-speed video-microscopy analysis (HSVM) and genetic testing. In our setting, TEM analysis was available only in a few cases because of logistic difficulties.

ERS guidelines [6] were followed to classify patients as: confirmed PCD (suggestive clinical history with hallmark ciliary ultrastructure defects assessed by TEM and/or presence of pathogenic bi-allelic variants in PCD-associated genes); highly likely PCD (suggestive history with very low nNO and HSVM findings consistently suggestive of PCD after repeated analysis or cell culture); or highly unlikely PCD (weak clinical history, normal nNO and normal HSVM).

Nasal nitric oxide (nNO) measurements were performed using CLD 88sp NO-analyzer (ECO MEDICS, AG, Duerten, Switzerland) according to ERS guidelines [13].

Genetic testing was performed with a high-throughput 44 PCD gene panel using the SeqCap EZ Technology (Roche Nimblegen, Pleasanton, CA, USA) as previously described [14]. Genetic results for most of the patients included in this study have been previously published [14].

HSVM was performed to study ciliary beat frequency (CBF) and ciliary beat pattern (CBP) (local normal values: CBF 8.7–18.8 Hz; CBP ≤20% dyskinetic ciliated cells). Nasal respiratory epithelia were sampled at the inferior nasal meatus with a 2 mm diameter brush submerged in HEPES-supplemented Medium199. A minimum of ten lateral strips with 10 cells each and two overhead axes were captured at 37 ºC with an optical microscope coupled to a high-speed video camera (MotionPro^®^ X4, IDT, CA, USA) using MotionPro^®^ X4 software [15].

### 2.3. Immunofluorescence Technique and Analysis

Nasal-brush respiratory epithelial samples were spread or dropped, air dried and stored at −80 °C until use. Cells were fixed with 4% PFA for 15 min at room temperature (RT), washed 4 times with 1xPBS, permeabilized with 0.2% TritonX100 for 10 min at RT and blocked with 1% fat-free skim milk in PBS overnight at +4 °C to avoid nonspecific binding. Samples were incubated with primary antibodies (all Sigma Aldrich, St. Louis, MO, USA) for 4 h at RT using the following dilutions in 1% skim milk: anti-DNAH5 antibody 1:200, anti-DNALI1 1:100, anti-GAS8 1:200, anti-RSPH4A 1:200 and anti-RSPH9 1:70.

We washed 5 times with 1xPBS at RT (2 washes of 10 min), and all slides were incubated for 45 min at RT with 1:2500 anti-acetylated tubulin antibody (Sigma Aldrich) for cilia localization. After 5 more washes with 1xPBS at RT (2 washes of 10 min), cells were incubated for 30 min at RT with secondary monoclonal anti-rabbit Alexa Fluor 594 and anti-mouse Alexa Fluor 488 antibodies (Thermo Fisher, Waltham, MA, USA). Nuclei DNA was stained with Prolong antifade DAPI (Thermo Fisher).

Slides were analyzed using a fluorescence microscope at X100 magnification. A minimum of ten cells were analyzed for each target protein. The results were considered: (1) normal or present when the protein was present in 8 or more cells, (2) absent or aberrant when the protein was completely absent in the ciliary axoneme or had an abnormal distribution in 8 or more cells, (3) inconclusive when results differed from previously described ones, and (4) insufficient when less than ten cells were observed. In inconclusive or insufficient cases, IF was repeated when possible, following recommendations by Shoemark et al. [5].

Patients were analyzed using antibodies against component proteins for the different structures of the ciliary axoneme: DNAH5 (an ODA component), DNALI1 (an IDA component), GAS8 (a nexin-dynein regulatory complex component) and radial spoke head components RSPH4A (42 patients), RSPH9 (31 patients) or both (1 patient). When this analysis was designed, we specifically chose and optimized these commercial antibodies to detect most cilia defects, following Shoemark et al. [5] and expert recommendations. Anti-acetylated tubulin antibody was used to localize the microtubular doublet (protein location shown in Figure 1).

### 2.4. Data Analysis

Confirmed and highly likely PCD cases were considered as positive for calculation of sensitivity and specificity. Data were analyzed by using MedCalc Statistical Software version 19.5.3 (MedCalc Software bvba, Ostend, Belgium).

## 3. Results

Immunofluorescence analysis was performed in a cohort of 74 PCD-suspected patients. Sixty-six percent of patients included in this study were <18 years old (49/74) and the mean age at study was 17.9 years (range 1–63).

After PCD diagnostic evaluation, 25 patients were considered as confirmed PCD, 25 as highly likely and 24 as highly unlikely PCD (Table 1).

IF was technically evaluable for all tested antibodies in 68 patients (91.9%), whereas in six, the results were inconclusive/insufficient (8.1%) (Table 1). IF analysis demonstrated an absence or aberrant localization of one or more proteins in the ciliary axoneme in 33 cases: 15 patients presented absent DNAH5; 3 a proximal localization of DNAH5 in ciliary axoneme; 3 patients exhibited absent DNAH5 and DNALI1; 7 showed absent DNALI1 and a cytoplasmatic localization of GAS8; 1 patient presented absent GAS8; 3 showed absent RSPH9; and 1 had absent RSPH4A (Table 1). Figure 2 shows examples of absence/aberrant localization of IF markers in four patients with confirmed PCD.

To evaluate the usefulness of IF analysis as a tool for PCD diagnosis, IF results were compared with those obtained from our PCD gene panel and HSVM analyses (Table 2). The 33 patients with aberrant/absent ciliary axoneme proteins had been diagnosed with confirmed or highly likely PCD. All of them presented a concordant abnormal HSVM and 21 of them presented likely pathogenic variants in PCD-related genes (Table 2). The relation of these abnormal IF results with clinical characteristics and other PCD diagnostic tests for each patient is shown in Appendix A.

Thirty-five patients had normal distribution or presence of all IF antibodies. The clinical characteristics and results of PCD diagnostic tests for each patient with normal IF are presented in Appendix A. We confirmed PCD in three of these patients because they presented likely pathogenic variants in *DNAH11* and hyperkinetic stiff cilia by HSVM (Table 2 and Appendix A). Another patient presented two variants in *SPAG1*, but there was no concordance with HSVM and IF results (Appendix A). Another 11 patients presented normal IF, but abnormal HSVM results together with typical PCD symptoms, so they were considered highly likely PCD (Appendix A). Thus far, we have not detected any likely pathogenic genetic variants in these patients. Finally, 20 out of the 33 patients with normal IF were considered highly unlikely to have PCD because of weak clinical history, normal or mild HSVM results and/or negative genetics (Appendix A).

It should be mentioned that six (8.1%) samples were not technically evaluable by our IF panel: two cases lacked enough cells to analyze and were considered insufficient; two cases remained inconclusive for one or more antibodies; and two cases resulted in being insufficient for some markers and inconclusive for others (Appendix A). Only one patient with insufficient and inconclusive IF sample was diagnosed with PCD because of presenting stiff cilia by HSVM and likely pathogenic variants in *CCDC39*. Among the other cases, four were finally regarded as having no PCD or unlikely to have PCD because of normal or mild HSVM results (Appendix A).

Considering all the previous results, IF analysis as a diagnostic test for PCD had a sensitivity of 68.8% (CI 95% 53.7–81.3%) and a specificity of 100% (CI 95% 83.2–100%). In our laboratory, the PCD prevalence (confirmed or highly likely cases) of patients referred for clinical suspicion in the last 4 years was 27.4% (non-published data). Assuming this prevalence, IF positive predictive value would be 100% and the negative predictive value 89.4% (CI 95% 84.8–92.8%).

## 4. Discussion

In this study, we have explored the diagnostic utility of an immunofluorescence panel of commercial antibodies in 74 patients with clinical suspicion of PCD. A technical evaluable result was possible in 91.9% of cases (Table 1). IF evidenced a protein defect in 44.6% of analyzed patients, all with confirmed or highly likely PCD (Table 1). A normal IF result (47.3% of cases) was seen not only in all non-PCD patients, but also in some patients with confirmed or highly likely PCD. This means that IF detected ciliary structural defects in 68.8% of confirmed or highly likely PCD patients. In our population, IF has shown the highest positive predictive value (a positive value is consistent with a PCD diagnosis) of 100% and a negative predictive value (a normal result may be seen not only in non-PCD patients but also in PCD patients) of 89.4%. Considering our low availability of TEM results, IF was a useful PCD diagnostic test, because it showed a sensitivity (68.8%) close to TEM studies, where 30% of all affected individuals had normal ciliary ultrastructure [16].

To our knowledge, only one previous study by Shoemark et al. [5] evaluated the accuracy of IF in PCD, concluding that IF and TEM have a similar diagnostic rate. Therefore, they proposed IF as a useful diagnostic tool when TEM equipment or expertise is not available, as IF is cheaper, easier to perform, requires more basic equipment and improves the turnaround time [5]. As TEM analysis was not available in most of our cases, we have confirmed that, under these circumstances, IF is a reliable test to study cilia structure. Furthermore, IF may be useful to confirm the results of other diagnostic tests like HSVM and genetics and guide new tests in those cases with absent or aberrant protein/s localization. In Shoemark’s study, IF failed to identify 12% of PCD cases [5], which is lower than the 31.3% of normal IF results that we found in our confirmed or highly likely PCD cases. This difference could be related to genetic differences between both series.

DNAH5 absence in ciliary axoneme correlated with immotile cilia by HSVM and variants in genes related to ODA defects concurring with other studies: *DNAH5* [17,18], *DNAI2* [19], *TTC25* [20] and *CCDC151* [21] (Table 2 and Appendix A). Moreover, we found proximal axonemal DNAH5 IF staining in three unrelated patients (Figure 2d) with mild clinical symptoms and subtle HSVM defects (mainly stiff and disorganized ciliary beat). One of them presented likely pathogenic variants in *DNAH9*, in concordance with recently published data [22,23] (Table 2 and Appendix A).

Some of the patients showed absence of both DNAH5 and DNALI1 (Figure 2a) also with completely immotile cilia. We could not find any candidate variant in these patients (Table 2 and Appendix A). These IF and HSVM results could be explained by genetic alterations in proteins involved in the assembly of both ODA and IDA [24,25,26,27,28,29,30,31,32,33,34,35] and further studies are warranted.

The patients with absent DNALI1 and abnormal localization (cytoplasmatic) of GAS8 (Figure 2b) had mainly stiff (reduced amplitude) and immotile cilia and likely pathogenic variants in *CCDC39* and *CCDC40* (Table 2 and Appendix A). These results are consistent with previous description of CCDC39 [36] and CCDC40 [37] as assembling factors of the IDA and the nexin–dynein regulatory complex structures [36,37,38].

Only one patient presented an absence of GAS8 in ciliary axoneme. This patient had hyperkinetic stiff cilia and respiratory symptoms beginning at neonatal age. These results could be explained by defects in the nexin–dynein regulatory complex (DRC) subunits, as previously described [39,40,41].

In our IF approach, radial spoke defects were first studied with the RSPH4A antibody and later with RSPH9. We decided to switch to RSPH9 because it is more informative for detecting all radial spoke head defects, and it has been recommended due to its reported absence from ciliary axonemes in radial spoke mutant cells [5,42]. In fact, one of our patients had normal RSPH4A, but absent RSPH9 (Figure 2c). Radial spoke defects in our patients were related to situs solitus and two different HSVM patterns: circular motion and stiff cilia, consistent with previously reported data (Table 2 and Appendix A) [42,43,44].

Our IF panel could not detect defects caused by *DNAH11* genetic variants in our patients, consistent with previously reported data [45]. For this reason, it would be interesting to include an anti-DNAH11 antibody in the IF panel, considering that it is commercially available, but it has not been optimized. As it happens with DNAH11, other ciliary proteins have been described to cause none or subtle ultrastructural defects: HYDIN [46], STK36 [47] and, most recently, SPEF2 [48]. STK36 has been described as a protein involved in the interaction between the central pair and the radial spoke [47]. HYDIN and SPEF2 have been functionally described to cause central pair defects in humans, and mutants of both proteins can be detected using antibodies against SPEF2 [48]. These ultrastructural defects could explain some of our normal IF results in highly likely PCD patients.

Some patients could not be resolved by IF as analysis was inconclusive and/or insufficient for some of the target proteins, requiring reevaluation of new brushing samples. Blood and mucus in the IF samples were found to be confounding factors in the analysis in a previous publication [5]. From our experience, we considered the slides with nasal brush sample prepared by dropping a better option than those by spreading. Slides with a dropped sample allowed a faster analysis due to having more cells in a smaller area. In addition, when the sample contained mucus, analysis was more complicated in spread samples, and usually there were not enough viable cells to complete the analysis.

The major limitation of the IF analysis is that, because of the use of primary antibodies directed to specific proteins, defects in unrelated proteins may be missed [5]. Moreover, patients with partial defects or missense mutations have been reported to have normal IF results [5], although we did not have any case with this particular observation in our cohort. As new genes and proteins related to PCD are discovered, the IF antibody panels may need to be revised and expanded in the future for an accurate diagnosis [49]. In fact, antibodies against a high number of ciliary proteins are already commercially available, although most of them have not yet been tested and/or validated for immunofluorescence or in human respiratory tissue [11]. For this reason, the optimization of antibodies in nasal brushing samples is difficult and time-consuming. Furthermore, from our experience, we have not even been able to properly optimize some commercially available antibodies, i.e., DNAH11. Further antibody optimization is necessary, and, as a matter of fact, Liu et al.’s extensive IF technical protocols may help with this [11]. Another pitfall is the lack of consensus regarding the performance of the IF technique and, more importantly, the agreement in the IF considerations when the analysis is performed. Currently, a consensus statement on IF, initiated during the European BEAT-PCD 2019, is on the way.

One important limitation of measuring the accuracy of IF for PCD diagnosis is the lack of a gold standard reference against which to measure it. In our study, we use the ERS task force criteria [6] assuming as a standard for comparison confirmed and highly likely PCD cases, with the added limitation of low availability of TEM analysis.

Moreover, the positive rate of our series was quite high (25 confirmed and 25 highly likely of 74 cases). This is related to a previous pre-screening for IF study of cases with more suggestive clinical symptoms.

Taking all into account, we propose a two-step IF analysis: a first panel with DNAH5, DNALI1, GAS8 and RSPH9 and, in cases with normal IF and consistent PCD suspicion (clinical symptoms and other techniques), a second IF round with antibodies against ciliary components associated with none or subtle ultrastructural defects: DNAH11 [45], STK36 [47] and SPEF2 [48]. Shoemark et al. [5] also recommended a first antibody panel with DNAH5, GAS8 and RSPH9 and omitted DNALI1 because its absence always coexists with an absence of DNAH5 or GAS8 [5]. This recommendation is supported by our cohort results, but as TEM is mostly unavailable and we are using IF results to clarify the genetics, we considered to maintain anti-DNALI1 antibody in our IF studies. Therefore, our proposed two-step IF analysis may be used in cases with non-available TEM. Alternatively, centers with available TEM analysis might use a first step IF panel with DNAH11, SPEF2, GAS8 and RSPH9 antibodies, omitting DNAH5 and DNALI1. For the translation to clinical diagnosis, Liu et al. also proposed a restricted 10-antibody panel (instead of 21) based on proteins which are non-detectable by TEM or those indirectly detecting mislocalization of other proteins (DNAH5, DNAH11, DNALI1, GAS8, CCDC65, RSPH4A, RSPH9, RPGR, OFD1 and SPEF2) [11]. Although a quantitative super-resolution imaging tool, such as the one proposed by Liu et al. [11], gives much more information and may solve so-called difficult or unsolved cases, it would be hard to implement in our clinical setting due to its high costs in time and personnel.

To conclude, the presented results confirm that IF is a reliable diagnosis technique for PCD (with a sensitivity of 68.8% and a specificity of 100%), even when TEM analysis is unavailable, although it cannot be used as a standalone test. Considering our results, we propose IF as a cheap, easy and widely available test to include in PCD diagnosis.

## Figures and Tables

**Figure 1 jcm-09-03603-f001:**
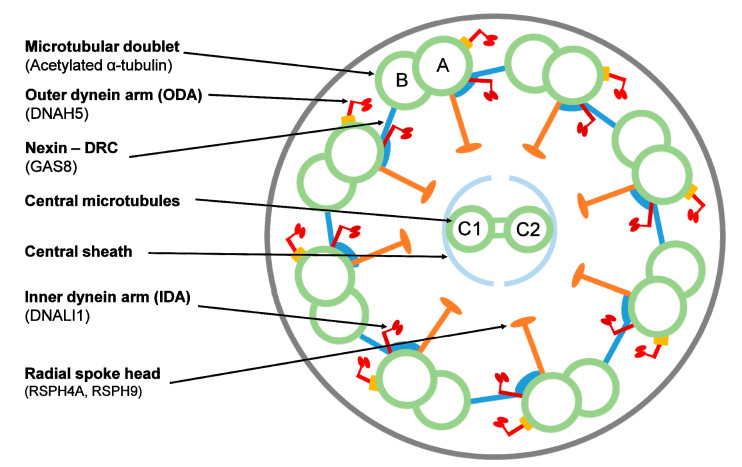
Ciliary axoneme in transverse section indicating the ultrastructural parts and the target proteins by immunofluorescence. Proteins are indicated in parentheses. DRC = dynein regulatory complex. A and B: outer microtubule doublets; C1 and C2: central pair.

**Figure 2 jcm-09-03603-f002:**
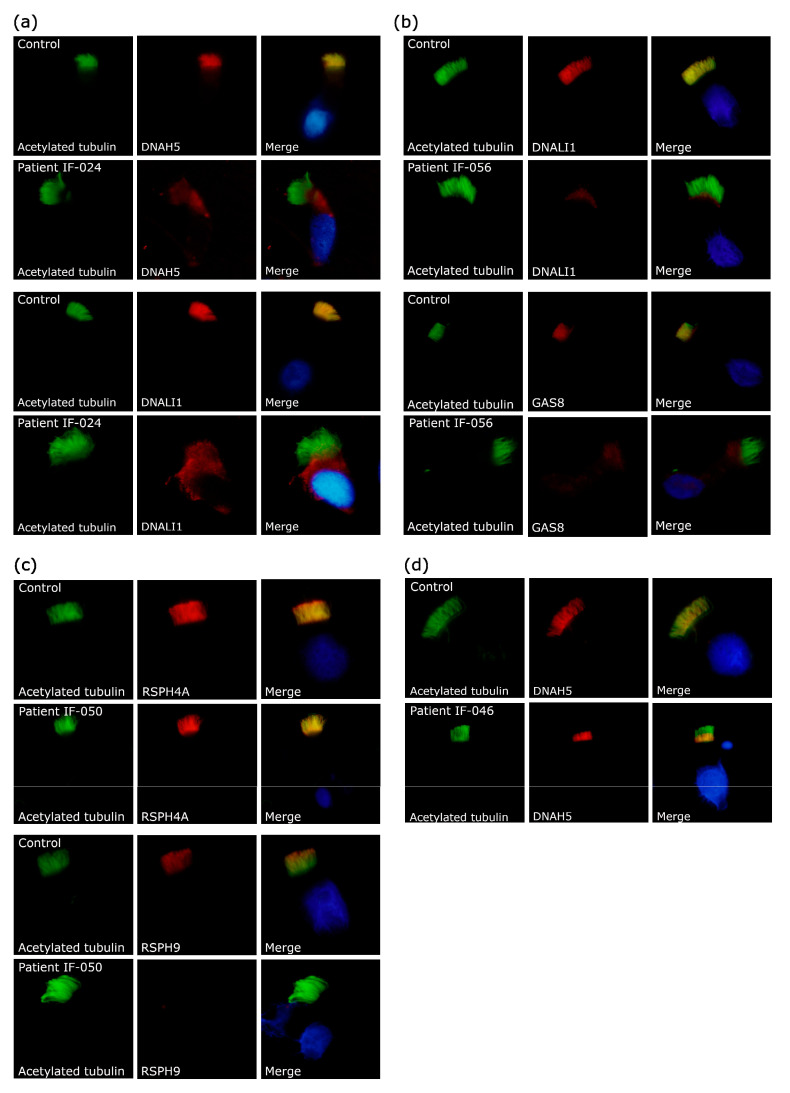
Example results of immunofluorescence technique in control subjects and patients with primary ciliary dyskinesia. The first column shows cilia by acetylated α-tubulin (green); the second column shows the protein of interest (red); and the third column shows the final merged image with the nuclei stained with DAPI (blue). (**a**) Patient IF-024 exhibited absent DNAH5 and DNALI1. (**b**) Patient IF-056 had absent DNALI1 and cytoplasmatic localization of GAS8. (**c**) Patient IF-050 showed a normal axonemal localization of RSPH4A and absent RSPH9. (**d**) Patient IF-046 presented a proximal localization of DNAH5.

**Table 1 jcm-09-03603-t001:** Immunofluorescence results from 74 primary ciliary dyskinesia (PCD) suspected patients related to the results of the PCD diagnostic evaluation.

		PCD Diagnostic Evaluation		
Immunofluorescence Test Outcome (*n* = 74)		Confirmed (*n* = 25)	Highly Likely (*n* = 25)	Highly Unlikely (*n* = 24)
Evaluable/Closed	68 (91.9%)	24	24	20
Normal results (all markers presents)	35 (47.3%)	3	12	20
Absent/aberrant results	33 (44.6%)	21	12	0
DNAH5 (-) (ODA)	15			
Proximal DNAH5 (ODA)	3			
DNAH5 (-), DNALI1 (-) (ODA+IDA)	3			
DNALI1 (-), GAS8 (-) (IDA+Nexin-DRC)	7			
GAS8 (-) (Nexin-DRC)	1			
RSPH9 (-) (Radial spoke)	3			
RSPH4A (-) (Radial spoke)	1			
Inconclusive/insufficient results	6 (8.1%)	1	1	4

(-): absent in ciliary axoneme; ODA: outer dynein arm; IDA: inner dynein arm; DRC: dynein regulatory complex.

**Table 2 jcm-09-03603-t002:** Correlation among immunofluorescence analysis, high-speed video-microscopy, genetics and clinical characteristics in our PCD patients.

IF Affected Markers (Ultrastructural Part)	#	HSVM	Genetics (#)	PCD Symptoms					
				Neonatal Distress	Upper Respiratory Tract	Lower Respiratory Tract	Bronchiectasis	Chronic Otitis or Hearing loss	Situs Abnormality
DNAH5 (ODA)	15	Completely immotile cilia or residual motility	*CCDC151* (1), *DNAH5* (5), *DNAI2* (4), *TTC25* (1), NA (4)	+/−	+	+	+/−	+/−	+/−
Proximal DNAH5 (ODA)	3	Subtle defects (stiff and disorganized ciliary beat)	*DNAH9* (1), Neg. (2)	-	+	+/−	+/−	+/−	+/−
DNAH5+DNALI1 (ODA+IDA)	3	Completely immotile cilia	Neg. (2), NA (1)	+	+	+	+	+	+/−
DNALI1+GAS8 (IDA+Nexin-DRC)	7	Mainly stiff cilia and immotile cilia	*CCDC39* (3), *CCDC40* (3), NA (1)	+/−	+	+/−	+/−	+/−	+/−
GAS8 (Nexin-DRC)	1	Hyperkinetic stiff cilia	NA (1)	+	+	+	+	+	−
RSPH4A or RSPH9 (Radial spoke)	4	Stiff and circular motion	*RSPH1* (1), *RSPH4A* (1), *RSPH9* (1), NA (1)	+/−	+	+/−	+/−	+/−	−
All markers present (normal result)	3	Hyperkinetic stiff cilia	*DNAH11* (3)	+/−	+	+/−	+/−	+/−	+/−

IF: immunofluorescence; #: number of patients; HSVM: high-speed video-microscopy; ODA: outer dynein arm; IDA: inner dynein arm; DRC: dynein regulatory complex; NA: not available data; Neg.: negative results; +: symptoms present in all patients; -: symptoms absent in all patients; +/-: symptoms present in some patients.

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
