# Peer review of "Immunofluorescence Analysis as a Diagnostic Tool in a Spanish Cohort of Patients with Suspected Primary Ciliary Dyskinesia"

_jcm, 2020, doi:10.3390/jcm9113603_

Round 1

Reviewer 1 Report

Baz-Redon et al. present a nice observational study, investigating the use of a panel of five commercially available antibodies to diagnose PCD. The study included 68 consecutive referrals for diagnostic testing; most participants had also had testing by nasal NO, high speed video analysis and genetics. TEM was not usually performed. There is limited information about use of IF, and although the results are confirmatory of previous literature, the data will be welcomed by people working in the field. The standard of English is very good and always understood; there are a few sentences which would benefit from editing.

I have some suggestions for improvements:

  1. Title: The study is a cohort referred for diagnostic testing, not a cohort with PCD.
  2. Abstract: It might be better to say ‘in the absence of TEM analysis’ rather than results.
  3. The introduction was clear, but I’d like the rationale for the choice of antibodies, either here or in the methods. E.g. why was DNAH11 and SPEF2 not included?
  4. Liu et al Sci Trans Med 2020 have also used IF to diagnose PCD, albeit with super resolution imaging. I’d suggest this is discussed alongside the Shoemark manuscript in the introduction and discussion; Liu et al used 21 commercially available antibodies and you could for example discuss some of the cases identified by their panel, but also the additional costs and person hours. Etc. Also the limitations and advantages of their methods using quantification.
  5. It would be helpful to also state the sensitivity and specificity against a definite positive diagnosis (in this study by TEM). Some discussion of the lack of a gold standard reference against which to measure the accuracy would be helpful.
  6. The positive rate is extremely high (25 confirmed and 25 highly likely of 74) in comparison to other diagnostic cohorts for PCD. This suggests some pre-screening of patients, or is there another explanation?
  7. Table 2: It would be helpful to indicate for each IF marker outcome, how many did not have genetics and how many had no mutation found.
  8. It is unusual that some patients with ODA defects had no lower (or upper) airway symptoms. It might be worth discussing this.
  9. It is worth highlighting that your suggestion for the 2step IF analysis with the proposed antibodies is in the context of no available TEM. A centre with TEM might reasonably use a first line panel comprising DNAH11, RSPH9, SPEF2 and GAS8, but not use DNAH5, DNALI1 for example.

Author Response

Authors: We thank the Reviewer for his/her considerations and comments.

1. Title: The study is a cohort referred for diagnostic testing, not a cohort with PCD.

 Authors: we have changed the title to Immunofluorescence analysis as a diagnostic tool in a Spanish cohort of patients with suspected primary ciliary dyskinesia (page 1, lines 2-4).

2. Abstract: It might be better to say ‘in the absence of TEM analysis’ rather than results.

 Authors: we have changed this sentence: “in the absence of transmission electron microscopy analysis” (page 2, lines 59-60).

 3. The introduction was clear, but I’d like the rationale for the choice of antibodies, either here or in the methods. E.g. why was DNAH11 and SPEF2 not included?

 Authors: we have added this clarification in the methods section (pages 4-5, lines 179-181): “Patients were analyzed using antibodies against component proteins for the different structures of the ciliary axoneme: DNAH5 (an ODA component), DNALI1 (an IDA component), GAS8 (a nexin-dynein regulatory complex component) and radial spoke head components RSPH4A (42 patients), RSPH9 (31 patients) or both (1 patient). When this analysis was designed, we specifically chose and optimized these commercial antibodies to detect most cilia defects, following Shoemark et al. [5] and expert recommendations.”

4. Liu et al Sci Trans Med 2020 have also used IF to diagnose PCD, albeit with super resolution imaging. I’d suggest this is discussed alongside the Shoemark manuscript in the introduction and discussion; Liu et al used 21 commercially available antibodies and you could for example discuss some of the cases identified by their panel, but also the additional costs and person hours. Etc. Also the limitations and advantages of their methods using quantification.

 Authors: thank you for these data. Our apologies. We added some ideas and information from these data through the manuscript:

 - in the introduction section (page 3, line, 110-115): “To our knowledge, this is the only study evaluating the accuracy of conventional IF for the diagnosis of PCD. In a recent study, Liu et al. presented a quantitative super-resolution imaging workflow for the detection of cilia defects thanks to the validation of 21 commercially available IF antibodies. Molecular defects using this super-resolution imaging toolbox were described in 31 clinical PCD cases, including patients with negative TEM results and/or with genetic variants of uncertain significance (VUS) [11]

 - and in the discussion section (page 9, lines 312: “although most of them have not yet been tested and/or validated” and 316-317: “Furthermore, from our experience, we have not even been able to properly optimize some commercially available antibodies, i.e. DNAH11. Further antibody optimization is necessary, and, as a matter of fact, Liu et al. extensive IF technical protocols may help on this [11]”,

-  and page 10, lines 338-344:” For the translation to clinical diagnosis, Liu et al. also proposed a restricted 10 antibodies panel (instead of 21) based on proteins non-detectable by TEM or those indirectly detecting other protein mislocalization (DNAH5, DNAH11, DNALI1, GAS8, CCDC65, RSPH4A, RSPH9, RPGR, OFD1, and SPEF2) [11]. Although a quantitative super-resolution imaging tool, such as the one proposed by Liu et al. [11], gives much more information and may solve so-called difficult or unsolved cases, it would be hard to implement in our clinical setting due to its high costs in time and personnel.”.

 - We also added this reference (page 3, line 115). Reference Liu et al. is now reference [11] and the correspondent changes in reference numbering have been made.

5. It would be helpful to also state the sensitivity and specificity against a definite positive diagnosis (in this study by TEM). Some discussion of the lack of a gold standard reference against which to measure the accuracy would be helpful.

 Authors: Thank you for the comment. We have added the following sentence in the Discussion section, line 321-324. “One important limitation about measuring the accuracy of IF for PCD diagnosis is the lack of a gold standard reference against which to measure it. In our study we use the ERS task force criteria [6] assuming as a standard for comparison confirmed and highly likely PCD cases, with the added limitation of low availability of TEM analysis.”  

6. positive rate is extremely high (25 confirmed and 25 highly likely of 74) in comparison to other diagnostic cohorts for PCD. This suggests some pre-screening of patients, or is there another explanation?

      Authors: Thank you for the comment. We have added the following sentence in the Discussion section, line 325-327. “Also, the positive rate of our series was quite high (25 confirmed and 25 highly likely of 74 cases). This is related to a previous pre-screening for IF study of cases with more suggestive clinical symptoms.” 

7. Table 2: It would be helpful to indicate for each IF marker outcome, how many did not have genetics and how many had no mutation found.

Authors: thank you for the suggestion. We added this information in the genetics results in Table 2 (page 7).

8. It is unusual that some patients with ODA defects had no lower (or upper) airway symptoms. It might be worth discussing this.

Authors: Thank you for this comment. We have reviewed our data and notice that there was an unintentional mistake in the table. All patients with ODA defects had both lower an upper airway symptoms. We have corrected the mistake in Table 2 (page 7).  

9. It is worth highlighting that your suggestion for the 2step IF analysis with the proposed antibodies is in the context of no available TEM. A centre with TEM might reasonably use a first line panel comprising DNAH11, RSPH9, SPEF2 and GAS8, but not use DNAH5, DNALI1 for example.

Authors: thank you for the suggestion. We added this information (page 10, lines 335-338): “Therefore, our proposed two-step IF analysis may be used in cases with non available TEM. Alternatively, centers with available TEM analysis might use a first step IF panel with DNAH11, SPEF2, GAS8 and RSPH9 antibodies omitting DNAH5 and DNALI1.

Reviewer 2 Report

I would like to congratulate the authors on this well written article describing the use of a panel of antibodies to diagnose PCD by IF. 

It is a ‘ really world’ study and as such some diagnostic information is missing for some patients, such as genetics. TEM is absent for most subjects also. However these constraints of the study are accurately described and addressed in the discussion. 

The study is very similar to that of Shoemark et al and reconfirms the findings of that study. This represents an important addition to the evidence base for use of IF as a diagnostic test and will hopefully be useful in the planning of the next set of diagnostic guidelines for PCD.

There are also weaknesses with the IF technique in that some cases of Primary Ciliary Dyskinesia will be missed. The authors address these well within the discussion. 

Author Response

We thank the rewiever for he/her comments.

We appreciate that he/she did not ask for any specific response

Please, see the attached file
